# What Determines the Entrepreneurial Intentions of Highly-Skilled Women with Refugee Experience? An Empirical Analysis in the Context of Sweden

**Nina Lazarczyk-Bilal * and Beata Glinka** 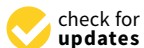

Faculty of Management, University of Warsaw, 02-678 Warsaw, Poland; b.glinka@uw.edu.pl
* Correspondence: nlazarczyk@wz.uw.edu.pl

**Abstract:** One of the main challenges faced by refugee hosting states is the labour market integration of newcomers, which can be achieved to some extent through the creation of small businesses. This paper analyses the individual level determinants of the entrepreneurial intentions of highly-skilled women with refugee experience. The study adds a new perspective to the conversation about highly-skilled migrant women analysed so far, mostly as family reunion migrants joining economic migrants. It also contributes to the relatively new research on refugee entrepreneurship by adopting an unusual perspective for looking at highly skilled women. The empirical analysis embedded in the context of Sweden is two-fold. First, it is done in SPSS on the sample ($N = 98$) drawn from the 2017 Swedish Invandrarindex data set with the use of binary logistic regression. Second, the findings from the quantitative analysis are nuanced with the analysis of two case studies based on SSI with Syrian women having refugee experience. The results show that the gender variable does not predict the effect on entrepreneurial intentions. The findings confirm the importance of previous self-employment and leadership experience and indicate the potential importance of entrepreneurial role models, the cultural aspect of entrepreneurial intentions and the role of an encouraging environment in the host country.

**Keywords:** highly-skilled migrant women; gender; refugee experience; entrepreneurial intentions; refugee entrepreneurship; Syria; Sweden; logistic regression; case study



## 1. Introduction

The global number of refugees, people who are forced to leave their country because of war or fear of persecution, has reached approximately 33.8 million (UNHCR 2020). Education is one of the key determinants in shaping the career trajectory of refugees. Highly skilled refugees are less likely than the non-qualified to suffer from unemployment and they are more likely to start a company after gaining their first work experience in the local labour market (Backman et al. 2020). Research reveals that the refugee gap, which stands for the long-term inactivity refugees often suffer from as a result of long asylum procedures, shrinks faster in the case of refugees with tertiary education (Bakker et al. 2017; Bygnes 2019; Dustmann et al. 2017). Nonetheless, even possessing high qualifications does not necessarily result in the long-term employment of refugees (Backman et al. 2020). First, as a consequence of conflict-related damages or a dangerous journey, refugees often lose their diplomas, which results in problems with the recognition of their qualifications in the host country (Alrawadieh et al. 2018; Bucken-Knapp et al. 2018; Colic-Peisker and Tilbury 2006; Wauters and Lambrecht 2006). Second, highly skilled refugees, often doctors and teachers, face the high risk of "deprofessionalisation" in the host country as a result of national entry requirements into the specific profession, no access to local professional networks and discrimination (Smyth 2010; Piętka-Nykaza 2015).

The study of highly-skilled refugees becomes further interesting in the context of the dubious narrative about economic migrants and refugees. On one hand, they are often

portrayed as a threat in the public discourse, e.g., it is broadly understood that migrants will take away jobs from the low-skilled population. On the other hand, a neoliberal narrative argues that the reception of highly-skilled refugees may be beneficial for the hosting economies in the EU market (Saner et al. 2019). In line with other authors, we argue that refugees must not be seen as assets, but diverse people searching for a safe haven due to war or fear of persecution (Ganassin and Young 2020; Sontag 2018; Heilbrunn and Iannone 2020). One of the main challenges faced by the refugee hosting states is to give newcomers financial independence and agency (Gold 1988; Ganassin and Young 2020). The labour market inclusion, which can be done either via employment or creation of business start-ups, is substantial (Predojevic-Despic and Lukic 2018; Kooli and Muftah 2020). Thus, we want to examine the influence of individual level determinants of the entrepreneurial intentions of highly-skilled refugees, women in particular.

Gender is definitely a key variable in determining the job-related choices of highly skilled migrant women (Grigoleit-Richter 2017). Research shows that some highly-skilled migrant women often plan to work in female-dominated sectors, which do not necessarily require a high level of qualifications (Zybura et al. 2018; Kooli and Muftah 2020). The traditional perception of women as mother and household keeper often reinforces the scenario in which the migrant woman does not enter the labour market and the man is a jobholder (Aure 2013). This may be common especially among migrants coming from societies with traditionally perceived gender norms (Abadli et al. 2020). The literature points out that migrant women are exposed to larger obstacles than men when starting a company, collecting funds and taking advantage of networks, which results in a lower rate of female-immigrant-owned companies (Brieger and Gielnik 2020).

Our goal is to analyse the individual level conditions under which highly-skilled women with refugee experience are more likely to have entrepreneurial intentions. Although we do a comparative study between women and men we address the research question: Under what conditions are highly-skilled women with refugee experience likely to have entrepreneurial intentions? To conduct our analysis, we draw upon the human capital theory, which focuses on individual aspects such as gender, age, level of education, qualifications and work experience (Vinogradov and Kolvereid 2007; Bach and Carroll-Seguin 1986; Kooli and Muftah 2020). Based on the literature, we define highly-skilled individuals as those who hold a university degree and declare to have a special education in a certain field (Chaloff and Lemaitre 2009; Lange et al. 2020).

By examining the determinants of the entrepreneurial intentions of highly-skilled women with refugee experience, we contribute to the intersection of three underdeveloped strands of literature, i.e., entrepreneurial intentions, refugee entrepreneurship and highly-skilled migrant women (Heilbrunn and Iannone 2020). Our study adds a new perspective of refugee experience to the conversation about highly-skilled migrant women analysed so far mostly as voluntary migrants joining, on the basis of the EU family reunification law, their family members with the status of economic migrants (Iredale 2005; Aure 2013; Mozetič 2018). Also, our paper contributes to the relatively new refugee entrepreneurship research. We adopt an unusual angle to examine the determinants of the entrepreneurial intentions of women with refugee experience, not just across gender, nationality or ethnicity, but looking through the lens of their qualifications and gender (Lange et al. 2020; Mozetič 2018). Last but not least, the existing studies both on refugee entrepreneurship and highly-skilled refugees are mostly based on qualitative analyses (Smyth 2010; Willott and Stevenson 2013; Psoinos 2007; Piętka-Nykaza 2015), which shows the need to conduct more quantitative research in this field (Heilbrunn and Iannone 2020). Our findings overall confirm the importance of previous self-employment, leadership qualities, occupational choices, entrepreneurial role models and the cultural aspect of entrepreneurial intentions.

We have structured this paper as follows. In the next section we discuss the literature review, focusing on the determinants of entrepreneurial intentions in the case of refugees. Then, we present the methods, data source, descriptive statistics and results obtained from the quantitative study. In Section 4 we include additional analyses based on the analysis of

two case studies. In the fifth part we discuss the results, and we provide the conclusions in the last section of the paper.

## 2. Entrepreneurial Intentions, Migration and Gender

The literature does not provide any straightforward answer to the question of what determines the entrepreneurial intentions of highly-skilled women with refugee experience. We see two reasons that could possibly explain the existence of the research gap we have identified. First of all, research on entrepreneurial intentions has been dominated by Ajzen's theory of planned behaviour (TPB) focusing solely on individual factors, and not group differences, in explaining entrepreneurial intentions (Liñán and Fayolle 2015). Second, studying refugees as an independent unit of analysis is a relatively new approach in academia. Until recently, research on refugee entrepreneurship has been treated as a part of the extensive field of migrant entrepreneurship (Glinka 2018). Only since 2015, when the EU witnessed a high influx of asylum-seekers, has refugee entrepreneurship started aspiring to become an independent area of study (Backman et al. 2020; Heilbrunn and Iannone 2020; Heilbrunn 2019; Alexandre et al. 2019; Alrawadieh et al. 2018; Shneikat and Alrawadieh 2019). Therefore, in the context of entrepreneurial intentions, research has mostly treated refugees as one broad category examined in opposition to economic migrants (Bevelander 2016; Lazarczyk-Bilal 2019).

In general, researchers argue that refugees encounter more barriers to start a company than economic migrants (Connor 2010; Sandberg et al. 2019; Wauters and Lambrecht 2006; Gold 1988). Unlike economic migrants who have been building long-term migration chains, refugees cannot benefit from strong social networks to start a company in the host country (Gold 1992, 1988; Wauters and Lambrecht 2006; Colic-Peisker and Tilbury 2006). Further, they can neither go back to their home country nor take advantage of transnational connections useful for potential business activity (Gold 1988; Portes et al. 2002). They are more likely than economic migrants to suffer from psychological trauma and they are highly uncertain about their future stay in the host country (Fuller-Love et al. 2006; Gold 1988), which may affect their self-employment propensity. Refugees also mention, much more often than economic migrants, such obstacles to start a company in the host country as lack of investment capital, complex bureaucracy and lack of self-confidence (Connor 2010; Wauters and Lambrecht 2006).

In Sweden, research findings suggest that migrants overall, refugees included, turn to self-employment because they cannot get any other job, for example, migrants from non-Western countries are excessively represented in the group of self-employed in Denmark and Sweden, and their income is lower from that of employed migrants (Andersson and Wadensjö 2004). In line with the disadvantage theory, research explains that migrants are usually at a disadvantaged position in the labour market being less educated and facing a language barrier. Moreover, the economic situation, the relatively higher number of incoming refugees in comparison to the past and discrimination in the host country do not make it easier for newcomers to find a job (Bevelander and Pendakur 2012).

Building upon previous research, we would like to move one step further in the analysis of entrepreneurial intentions and take into account the differences among refugees, who naturally do not constitute a homogenous group. In our analysis we focus on highly-skilled women with refugee experience. We stress the refugee experience, which is similar for asylum-seekers and their reunited family members, as all are fleeing from lethal danger in their home country. Hence, although we do acknowledge that legal migration status entails various consequences for newcomers in the host country, we treat asylum-seekers and their reunited family members as one category. Additionally, with the intention to stress the diversity of refugees, we use in our analysis the terms people or individuals with refugee experience to emphasize that having a refugee experience is only one of the multiple identities among e.g., being a parent, sibling, teacher, shopkeeper, Christian, Muslim, liberal, conservative, atheist or national of a certain country, etc. (Ganassin and Young 2020; Mozetič 2018). We draw upon the human capital theory to focus on individual

aspects determining entrepreneurial intentions, which explains our choice of predictor variables, i.e., gender, previous self-employment experience, type of occupation (prone to employment or self-employment), leadership aspiration and age (Vinogradov and Kolvereid 2007; Bach and Carroll-Seguin 1986; Kooli and Muftah 2020).

### 2.1. Gender

Since we are interested in the entrepreneurial intentions of highly-skilled women with refugee experience, gender is the most important dimension in our study. In line with previous research, we expect highly-skilled women with refugee experience to be less likely than their male counterparts to have entrepreneurial intentions in the host country (Andersson Joona and Wadensjö 2008; Brieger and Gielnik 2020). The literature indicates that migrant women are exposed to larger obstacles than men when setting up a business, collecting start-up capital and benefitting from networks, which results in a lower rate of female-immigrant-owned companies (Brieger and Gielnik 2020).

**Hypothesis 1 (H1).** *Highly-skilled women with refugee experience are less likely to have entrepreneurial intention in the host country than their male counterparts.*

Building upon the previous research, we believe that gender does not represent only one variable, but it creates a whole framework to study entrepreneurial intentions from an institutional, local and personal perspective (Brush et al. 2009; Bastian et al. 2018). Even when gender is studied at the individual level, as in this paper, one needs to bear in mind the broader context in which gender norms evolve and play out on a daily basis (Brush et al. 2009). Thus, in the case of women with refugee experience, it is so important to take into account the impact of both their home country culture as well as the host country environment. In the case of our analysis we discuss the transition of women from a gender-inegalitarian regime to a gender-egalitarian host country (Tahir 2020). On one hand, Syria has one of the lowest female self-employment rates in the world and a deeply gendered society with a cultural and institutional system that discourages women from engaging not only in entrepreneurship (Bastian and Zali 2016) but also in the labour market (Kooli and Muftah 2020). On the other hand, Sweden guarantees women's rights and a whole set of new liberal ideologies to interact with, which alone is not enough to affect a belief in conservative gender norms (Tahir 2020).

### 2.2. Previous Self-Employment

Second, we argue that previous self-employment experience increases the probability of entrepreneurial intentions among migrants in the host country (Alexandre et al. 2019; Kachkar 2019). For example, many Syrian refugees who had run their own ventures back home opened similar companies in Turkey (Demir 2018). Findings from Belgium also show the tendency that Syrian refugees with previous self-employment experience from their home country choose to start their own company in the host country (Wauters and Lambrecht 2008). Since running one's own company has been often a family tradition or a family business refugees are tempted to follow the entrepreneurial role models in the family (Wauters and Lambrecht 2006, 2008).

**Hypothesis 2a (H2a).** *Highly-skilled individuals with refugee experience having previous self-employment are more likely to have entrepreneurial intention than those without previous self-employment.*

**Hypothesis 2b (H2b).** *Highly-skilled women with refugee experience having previous self-employment are less likely to have entrepreneurial intention than their male counterparts.*



### 2.3. Employment Prone Occupation

Third, studies show that, in line with the social role theory, women and men opt for occupations that are traditionally assigned to their genders (Campion 2018). Thus, we argue that highly-skilled people with refugee experience having worked in more employment prone sectors—such as teaching or healthcare—are less likely to have entrepreneurial intentions. On the contrary, men are much more likely to be seen as prone to self-employment (Kõu and Bailey 2017).

**Hypothesis 3a (H3a).** *Highly-skilled individuals with refugee experience having employment prone occupation are less likely to have entrepreneurial intention than those with self-employment prone occupation.*

**Hypothesis 3b (H3b).** *Highly-skilled women with refugee experience having employment prone occupation are less likely to have entrepreneurial intention than their male counterparts.*

### 2.4. Leadership Interest

Fourth, we expect highly-skilled people having refugee experience with some leadership experience or leadership interest to be more likely to have entrepreneurial intention. According to Obschonka, leadership in sport may predict entrepreneurial intention (Obschonka 2016). We find it relevant in the case of individuals with refugee experience who demonstrate an interest in a leadership position in sport in the host country. It can also be seen as an indicator of proactivity or self-initiative, which are also important for entrepreneurial intentions (Obschonka et al. 2018).

**Hypothesis 4a (H4a).** *Highly-skilled individuals with refugee experience having some leadership aspiration in the host country are more likely to have entrepreneurial intention than those without it.*

**Hypothesis 4b (H4b).** *Highly-skilled women with refugee experience having some leadership aspiration in the host country are more likely to have entrepreneurial intention than those without it.*

### 2.5. Age

Last but not least, we assume that young people are more motivated than older people to start a company in the host country, and we hypothesise that young women are less likely to have entrepreneurial intention than their male peers.

**Hypothesis 5a (H5a).** *Younger highly-skilled individuals with refugee experience are more likely to have entrepreneurial intention than older individuals.*

**Hypothesis 5b (H5b).** *Younger highly-skilled women with refugee experience are less likely to have entrepreneurial intention than their male counterparts.*

**Hypothesis 5c (H5c).** *Younger highly-skilled women with refugee experience are less likely to have entrepreneurial intention than older highly-skilled men with refugee experience.*

### 3. Methods

We conducted a two-fold empirical analysis in the context of Sweden. First, we conducted a quantitative analysis with the use of the IBM SPSS 26 package program. We ran a binary logistic regression on the sample ($N = 98$) drawn from the 2017 Swedish Invandrarindex data set. It is recommended that logistic regression be used when the outcome variable is categorical, whereas predictor variables are either categorical or continuous (Field 2009). Since our outcome variable (Entrepreneurial intentions: 1 = Yes, 0 = No) is dichotomous, we applied binary logistic regression.

In the second step, we completed the obtained quantitative findings with a qualitative analysis of the two case studies based on independently collected material, i.e., interviews

with Syrian women having refugee experience. The role of the qualitative analysis is to nuance and put in context the quantitative findings (Gehman et al. 2018). Although the interviewees are not highly skilled, we argue that the material provides valuable insights about the life of Syrian women with refugee experience in Sweden. Coming from the same country and having refugee experience as a consequence of war, they may share many similar experiences and reflections about life in Sweden.

### 3.1. Data Source

The data source for the quantitative part comes from the "Invandrarindex ("Immigrants' index")—the new Swedes' voice". The data were collected anonymously in Sweden between 11 September 2017 and 15 October 2017 in 30 various municipalities across Sweden. The survey target group comprised 2526 migrants (refugees, economic and family reunion migrants) who arrived to Sweden in the second decade of 2000, aged between 18 and 70 years old, who participated in the state-sponsored language course, Swedish for Foreigners (Svenska för Invandrare, SFI). The respondents were given a multiple choice questionnaire of 157 questions. They filled it out through a web survey during an SFI class. They had 60 min to fill out the questionnaire. The survey was available in six languages, i.e., Swedish, English, Arabic, Dari, Somali and Tigrinya.

The case studies are based on two semi-structured interviews conducted in September 2019 in Sweden in Uppsala. Both of them had completed a high school education back home and do not fall into the category of highly-skilled migrants, which in our opinion nonetheless provides added value to our study. Although not highly-skilled, their personal stories shed light on the results of our empirical analysis and may be helpful to understand the entrepreneurial intentions of highly-skilled women. At the time of conducting the interviews they were unemployed. The interviewees were reached through a Swedish language school, which is a gathering place for many newcomers.

In Sweden, attending free Swedish classes is part of the mandatory training for asylum-seekers and one of the requirements to obtain state financial assistance. Thus, most of the newcomers with refugee experience go through the Swedish course. Both interviews from the article were held in English since the interviewees spoke English well. On one hand, this is a plus because there was no need for an interpreter and there was direct contact as well as understanding between the interviewer and interviewees. On the other hand, providing case studies based only on interviews with English speaking women poses a risk of a self-selected sample. However, in this paper the case studies are only supposed to illustrate and nuance the empirical analysis by highlighting the personal stories of women with refugee experience. It is also worth noting that foreigners in the school who spoke English were more open and eager to take part in the interview, whereas newcomers speaking solely Arabic or other foreign languages were not willing to establish any contact even to arrange future interviews with the participation of an interpreter.

### 3.2. Sample and Variables of Interest

In order to study the entrepreneurial intentions of highly-skilled refugee women, we selected a sample of 98 respondents. In order to delimit the sample we had to apply several criteria such as the lack of values missing on the crucial variables of our interest, refugee background of respondents and their tertiary level of education. We applied the nationality criterion reducing our sample solely to Syrian nationals to make sure that all the respondents in our sample possess refugee experience regardless of their legal entry status. Then, we have selected respondents who are university graduates and declare to have an acquired education in a certain field to obtain a sample of highly-skilled respondents. We have checked all the assumptions indispensable for logistic regression (Field 2009).

The outcome variable is a dichotomous variable based on the question Do you want to start your own company in Sweden? (1 = Yes, 2 = No, 3 = I don't know), which we have decided to recode into dummy variables (1 = Yes, 0 = No). Since we are interested in the individual level determinants of entrepreneurial intentions, we have included the following

predictor variables: Gender (1 = Yes, 0 = No), Previous self-employment (1 = Yes, 0 = No), Leadership aspiration in Sweden (1 = Yes, 0 = No) and Employment prone occupation (1 = Yes, 0 = No). We controlled for Age. Since we are interested in the influence of gender we also inserted three interactions between gender and other variables (Gender × Previous self-employment, Gender × Employment prone occupation as well as Gender × Age).

While most of the variables are self-explanatory, we should explain the variable Leadership experience in Sweden (1 = having some, 2 = interested in having it, 3 = not at all interested), which we have dummy coded (1 = Yes, 2 = No). It is based on the survey question Have you been active in sport as a leader, coach, board member or similar since you came to Sweden?.

The variable Employment prone occupation (1 = Yes, 0 = No) is a dummy variable in which 1 stands for professional experience in teaching or the healthcare sector, the most employment prone professions and at the same time the least prone to self-employment (Zybura et al. 2018). The value of 0 relates to all other occupations.

*3.3. Descriptive Statistics*

The selected sample consists of 98 highly-skilled Syrian respondents with refugee experience with a bigger share of men (59.2 percent) than women (40.8 percent) (Table 1). The average age of the respondents is 32.2 years old (median = 30, mode = 23). The most numerous age group is 25–34 years old (44.9 percent) and is followed by 35–44 years old (22.4 percent), below 24 years old (20.4 percent) and 45 or above (12.2. percent). More than one fifth of all the respondents have previous self-employment experience (21 percent), which means that they had run their own company prior to coming to Sweden. Just under one third (33.7 percent) have worked in employment prone professions, 8.2 percent have acquired some leadership experience in Sweden, 39.8 percent are interested in gaining leadership experience and 52 percent are not interested at all. When asked whether they want to start their own company in the host country, 36.7 percent of highly-skilled respondents with refugee experience answered Yes and 63.3 percent said No. All the respondents were unemployed at the time of filling out the survey.

The female respondents are on average 33.2 (median = 31.5, mode = 23) years old, whereas men are on average 31.5 (median = 29.5, mode = 23, 25, 30) years old. The majority of female respondents are between 34 and 45 years old (32.5 percent). The second-biggest age group among female respondents is between 25 and 34 (30 percent), the next is between 21 and 24 years old (22.5 percent) and the last one is 45 or older (15 percent). Twelve and a half percent of highly-skilled female respondents possess previous self-employment experience, which is a much lower number in reference to men (27.6 percent). More than half of female respondents have experience in employment prone professions (55 percent), whereas in case of male respondents the number is substantially lower (25.9 percent). Ten percent of female respondents have some leadership experience, 30 percent are interested in having it and 60 percent are not interested at all. Among men, 6.9 percent have some leadership experience, 46.6 percent are interested in acquiring some and 46.6 percent are not interested at all in leading a team. Slightly more female respondents (37.5 percent) than men (36.2 percent) want to start their own company in Sweden.

As we can see in Figure 1, most of the female respondents have an educational background in teaching (37.5 percent), administration, economics, law (25.9 percent) and healthcare (10 percent), whereas the majority of men have completed their education in administration, economics, law (25.9 percent), teaching (19 percent) and data or IT (13.8 percent) (detailed numbers are available in Appendix A in Table A1). Seventeen and a half percent percent of female and 17.2 percent of male respondents have education in fields other than the options available in the questionnaire.

**Table 1.** Descriptive statistics. Predictor variables and outcome variable based on the sample (*N* = 98) of highly-skilled respondents with refugee experience.

| Variable | | All | % | Women | % | Men | % |
|---|---|---|---|---|---|---|---|
| Gender | | | | | | | |
| | Women | 40 | 40.8 | 40 | 100 | 0 | 0 |
| | Men | 58 | 59.2 | 0 | 0 | 58 | 100 |
| Age (years) | | | | | | | |
| | 24 or below | 20 | 20.4 | 9 | 22.5 | 11 | 19 |
| | 25–34 | 44 | 44.9 | 12 | 30 | 32 | 55.2 |
| | 35–44 | 22 | 22.4 | 13 | 32.5 | 9 | 15.5 |
| | 45–55 | 12 | 12.2 | 6 | 15 | 6 | 10.3 |
| Previous self-employment | | | | | | | |
| | Yes | 21 | 21.4 | 5 | 12.5 | 16 | 27.6 |
| | No | 77 | 78.6 | 35 | 87.5 | 42 | 72.4 |
| Dependent prone profession | | | | | | | |
| | Yes | 33 | 33.7 | 19 | 47.5 | 14 | 24.1 |
| | No | 65 | 66.3 | 21 | 52.5 | 44 | 75.9 |
| Leadership experience in Sweden | | | | | | | |
| | Yes | 47 | 48 | 16 | 40 | 31 | 53.5 |
| | No | 51 | 52 | 24 | 60 | 27 | 46.5 |
| Entrepreneurial intention | | | | | | | |
| | Yes | 36 | 36.7 | 15 | 37.5 | 21 | 36.2 |
| | No | 62 | 63.3 | 25 | 62.5 | 37 | 63.8 |

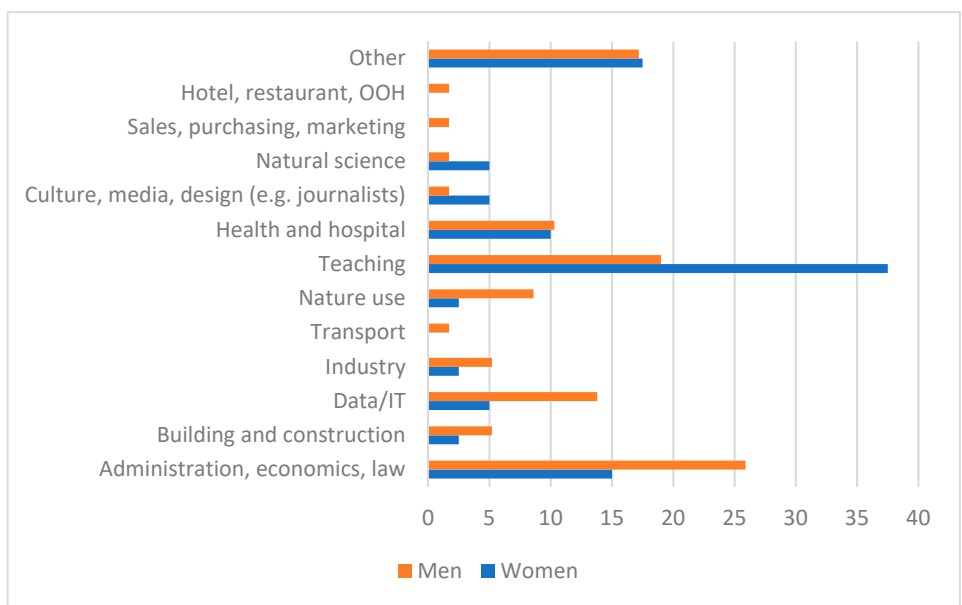

**Figure 1.** Educational background of highly-skilled men and women (*N* = 98) with refugee experience before coming to Sweden.

Figure 2 shows that women clearly differ from men in terms of their occupational choices when it comes to professional experience. The two dominant sectors among female respondents are teaching (45 percent) and healthcare (10 percent). The remaining female respondents have worked in various fields. The occupational choices of men in reference to job experience are much more diversified, i.e., sales, marketing, purchasing (20.7 percent), teaching (19 percent), industry (17.2 percent), administration, economics, law (15.3 percent), nature use (15.5. percent), building and construction (13.8 percent), data/IT (12.1 percent), health and hospital (6.9 percent) and others. Fifteen percent of female and 10.3 percent of male respondents have no professional experience.

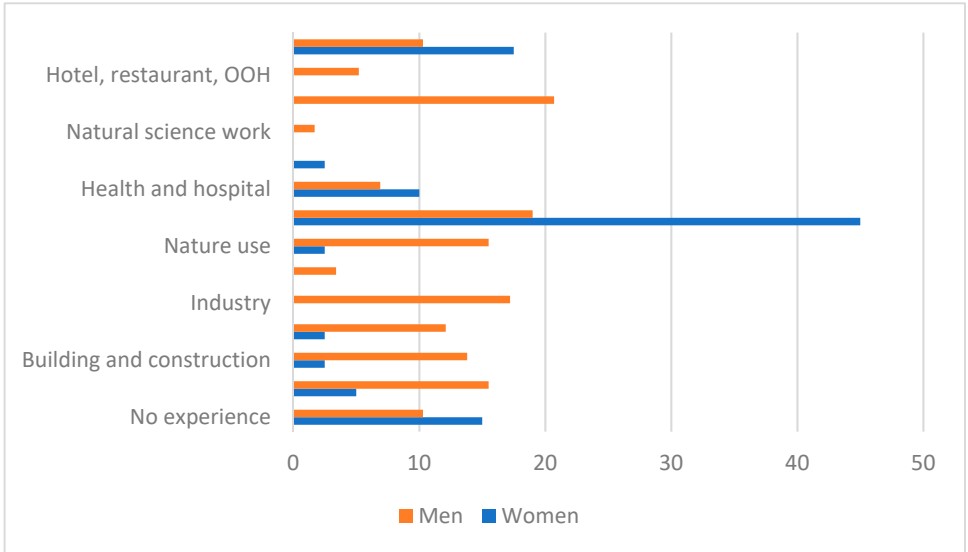

**Figure 2.** Work experience of highly-skilled men and women (*N* = 98) with refugee experience before coming to Sweden.

### 3.4. Binary Logistic Regression Model

In this section we present findings from the binary logistic regression in which we test the correlation between the outcome variable (entrepreneurial intention: 0 = No, 1 = Yes) and four predictor variables (gender, previous self-employment, leadership aspiration and employment prone occupation). We controlled for age. Additionally, we added four two-way interactions of gender with leadership aspiration, gender with previous self-employment, gender with employment prone profession and gender with age. Our model fulfills all the assumptions for the logistic regression (Field 2009).

Table 2 presents the results. The model fits the data well, as evidenced by the statistically significant value of the chi2 test ($p < 0.05$) and the statistically insignificant value of the Hosmer and Lemeshow test ($p > 0.05$). The model correctly classifies 69.4 percent of the respondents. The results show that previous self-employment and leadership experience variables predict the effect on entrepreneurial intentions. First, the respondents with previous self-employment experience are more likely to have entrepreneurial intentions. Previous self-employment significantly predicts whether highly-skilled individuals with refugee experience have the intention to start a company in the host country (b = 1.64, Wald $\chi 2$ (1) = 4.76, $p = 0.029$). The odds ratio indicates that as previous self-employment increases from No (0) to Yes (1), the change in the odds of having entrepreneurial intentions is 5.16. Second, the less the respondents are interested in gaining leadership experience, the less they are likely to have entrepreneurial intentions. In other words, the respondents with leadership experience or those interested in gaining it are more likely to have entrepreneurial intentions. Leadership experience significantly predicts whether highly-skilled individuals with refugee experience have intentions to start a company in the host country (b = −1.09, Wald $\chi 2$ (1) = 7.897, $p = 0.005$). The odds ratio indicates that as previous self-employment increases from No (0) to Yes (1), the change in the odds of having entrepreneurial intentions is 0.337. Gender, employment prone occupation, age and all four interactions were not statistically significant.

**Table 2.** Parameter estimates of the binary logistic regression model [1].

| Entrepreneurial Intentions | B | (SE) | *p*-Value | | Lower Bound | Odds Ratio | Upper Bound |
|---|---|---|---|---|---|---|---|
| Predictor variables | | | | | | | |
| Gender | −1.094 | 1.930 | 0.571 | | 0.008 | 0.335 | 14.720 |
| Previous self-employment | 1.641 | 0.753 | 0.029 | ** | 1.181 | 5.162 | 22.563 |
| Employment prone occupation | −3.478 | 1.847 | 0.060 | | 0.001 | 0.031 | 1.153 |
| Leadership experience | −1.086 | 0.387 | 0.005 | ** | 0.158 | 0.337 | 0.720 |
| Gender by age | −0.075 | 0.062 | 0.228 | | 0.509 | 0.928 | 42.769 |
| Gender by previous self-employment | −1.874 | 1.559 | 0.229 | | 0.007 | 0.154 | 3.259 |
| Gender by employment prone occupation | 1.541 | 1.130 | 0.173 | | 0.822 | 4.667 | 1.048 |
| Age | 0.162 | 0.103 | 0.114 | | 0.962 | 1.176 | 1.438 |
| Constant | 2.269 | 1.733 | 0.190 | | | 9.673 | |

[1] Note: R2 = 0.205 (Cox & Snell), 0.281 (Nagelkerke). Model χ2 (8) = 22.517, $p < 0.001$. **, $p < 0.01$.

## 4. Additional Analysis Based on Two Case Studies

To examine further the results of our quantitative study, we draw on the analysis of two case studies. The case studies are based on semi-structured interviews with two Syrian women having refugee experience and living in Sweden. The examination of case studies deepens the interpretation of our understanding of what are the individual level conditions that make highly-skilled women with refugee experience likely to have entrepreneurial intentions. Although the interviewees have not completed tertiary education we argue that the analysis of their stories can enhance the understanding of some of the obstacles and enablers highly-skilled women with refugee experience also face when they consider starting a business activity. We identify five main obstacles such as lack of previous self-employment, the tendency of women to lean towards employment prone occupations, lack of start-up capital, no knowledge or support of business incubators and uncertainty about the future. Nonetheless, we also identify several enablers of entrepreneurial intentions like the impact of entrepreneurial role models, home country entrepreneurial culture as well as the supportive role of the host country environment. In the following text we will first, briefly, present the interviewees, second, discuss the challenges and then the enablers of the entrepreneurial intentions of women with refugee experience.

The two interviewees are young women, 25 and 34 years old, who have completed secondary education back in their home country. They both have refugee experience since they arrived in the host country respectively in 2017 and 2018 as family reunion migrants to join their husbands who had sought asylum in Sweden. Nevertheless, these are not very similar cases, because their stance on entrepreneurial intentions is different. The first woman does declare having entrepreneurial intentions; however, it is a rather remote and vague plan since, as she notes, it takes time and start-up capital. She would like to start a company in Sweden with her husband either related to orthodontist products or a franchise of a Turkish store with clothes for Muslim women. She has pointed out "good money" and a more flexible working schedule relevant for work-life balance as the benefits of running one's own company. Nonetheless, she has stressed that her priority is to graduate from university. In Sweden she is planning to study either economics or orthodontics "something in the whole world it's working for me". The second interviewee, when asked at the beginning of the interview about her potential entrepreneurial plans, claimed she was not interested in entrepreneurship at all. However, later on it turned out that she had been actively supporting her husband in starting a company in Sweden. Since she spoke English well she has played a key role in getting information about starting a new business in the host country, finding and negotiating the conditions to rent a location for the business activity. Her husband, who had run with his brother a repair shop of electronic devices in Syria, wanted to continue the same business activity in the host country.

*4.1. Obstacles to Entrepreneurship*

Although one of the interviewees clearly admits having some entrepreneurial intentions, neither of them has previous self-employment experience. Both of them are only considering employment prone occupations such as a babysitter, teacher or caretaker. One of them commented on the issue of women's employment in Syria: "We don't have a problem with the job [of women], we have the problem with the type of the job. When you have a job, like, when you have your pharmacy, it's okay, the man doesn't say no, you can work. When you are a doctor, when you are a teacher, this is not problem at all. But other jobs, that's not good for them". In both cases the question of start-up capital is a challenging aspect of starting a company. The second interviewee, who together with her husband has already started the process of establishing a company in Sweden, has talked about the difficulty in raising start-up capital. Taking a loan from a bank requires long-term employment, which unemployed newcomers do not have. Needless to say, she has also mentioned that having some initial capital is not a solution, because the moment one would like to transfer money to the bank account, the bank deducts 33 percent of tax. In the case of refugees whose social networks and family ties are disrupted in the host country, borrowing money from relatives or friends is not an option: "[W]e all came from the same situation. War, our money was taken. You know, our things, our jewelry were robbed from us. So, you can't find in your family or your friends someone who has money to give you, for like a year. They can't, they can't do that". The interviewee has also mentioned the challenging bureaucracy, expensive rental of local space and most likely some form of discrimination, which hinder further the venture creation process for people with refugee experience. When her husband eventually found an affordable place for business activity and they were about to sign the contract with an owner, the owner changed his mind and decided to sell it although he was very clear about renting it out from the beginning: "We found a place a few weeks ago and the owner didn't ask much. But when we say okay, let's do the contract, it's "No, I want to sell the shop" ( . . . ) So we dropped the idea of opening in Uppsala, we are going to another city to try that". It also turned out that they could not rely on the support of local business incubators. The interviewee together with her husband have established contact with one of the organizations endorsing start-ups, but they told them: "Okay, find the location and give us the idea, we will make our study. Then we will see if we will help you or not. So we are trying to find that location". The interviewee and her husband were struggling to reach at least the stage at which incubators start providing support to entrepreneurs. The first interviewee did not even have any knowledge of such organizations.

The attitude towards language proficiency in Swedish was a bit surprising since neither interviewee raised it as a strong obstacle. They both believe it is necessary to learn the host country language, but have not mentioned it as a hindrance in the venture creation process. The analysis of the interview with the second woman has provided some mixed findings. On one hand, she explained that in her opinion the knowledge of the local language is not a barrier. She argued that since she speaks fluent English and her husband speaks Swedish they can easily communicate with other locals. On the other hand, her knowledge of English was crucial to negotiating the rental details, because her husband's Swedish skills were not sufficient to discuss the rental conditions. This situation indicates that sometimes good-enough knowledge of a language is not enough when it comes to negotiating business- or legal-related matters. Thus, an excellent level of either Swedish or English is crucial for nascent entrepreneurs to take action in the host country. Additionally, she has explicitly pointed out language as an obstacle while talking about the job search process: "How can I say this... most of Swedish people want to be perfect and Swedish, writing, reading, etc., so you don't have a chance here. You go to immigrants employ and they want to use you, work for so many hours, low money". Last but not least, the concept of uncertainty emerged from both interviews although it was not directly mentioned in the context of entrepreneurial intentions. Both interviewees have been granted only short-term temporary residence permits, for 1.5 and 2 years, which is a source of uncertainty and

anxiety about their future stay in the host country. The uncertainty about the prolongation of the residence permits may discourage individuals with refugee experience from starting a company. The venture creation process may take at least one or two years to establish a new firm in the market so it starts bringing profits. As in the case of the first interview, the young woman desires to pursue her education at the university and she would like to set up a business only once she becomes settled in the host country. What is more, she is not sure whether she and her family will stay in Sweden after obtaining Swedish citizenship. She has explained that when wearing a headscarf she feels like an outsider and often does not feel welcome in Sweden.

*4.2. Enablers of Entrepreneurial Intentions*

After examining the challenges women with refugee experience see or face in the business start-up process, we now move on to the analysis of the main enablers of entrepreneurial intentions. We identify three main aspects that encourage women with refugee experience to engage in the entrepreneurial process or enter the host country labour market, i.e., the impact of an entrepreneurial role model, Syrian entrepreneurial culture and the supportive environment in the host country. As it was mentioned above, neither interviewee has experience of previous self-employment and yet, one of them has been considering future venture creation and the other has been engaged in the business start-up process. Although they have not had their own company in the past, they both had been surrounded by their family members, father, husband and uncles, running their own shops back in Syria. Importantly, in both cases the interviewed women were not acting or planning alone, but together with their husbands. Both interviewees have also highlighted the role of Syrian culture, which incites people to run their own company. According to the interviewees, self-employment is the best way to make a profit, and thus, it is more prestigious than having a government job. In short, "it's a great thing [to have your own shop]".

Finally, both interviewees have discussed the supportive role of the environment in the host country, i.e., working women being a norm in the society, the importance of women's rights and the availability of childcare facilities. On one hand, the first interviewee has shared that she would like to work because it is a norm for women there to be professionally active. She also added: "I don't like to stay all the time with them because I need something to do for myself". On the other hand, the second interviewee has highlighted the importance of women's rights, which play a key role in women's life and career: "I have here, in Sweden, what I didn't have in my country, that woman's right. I struggled in Syria to have my rights. Here, they are delivered to you. You have them. They help you to do anything you want". In Syria she was forced to leave her modelling career: "I stopped working the day I got married, because there, it's not usual for the girl to work as a model. So his family said 'She has to stop', so the neighbours and people you know, stop speaking bad about us. So I stopped". The described situation is an example of the emancipatory dimension of refuge. Paradoxically, women with refugee experience who flee their home country with conservative gender norms regain their rights in the liberal democratic host country.

Last but not least, both interviewees take advantage of the childcare facilities. The first interviewee sends her daughter to a daycare centre, so she can take part in state-sponsored Swedish classes. The second interviewee has emphasized the importance of the availability of childcare facilities. These institutions support mothers in taking care of their children, which gives women a chance to pursue their own career: "You have kindergarten and after that, if you are working, you can put him for free with the teachers. So something might look small, but I think it's huge for women, especially women who came from Middle East. There, you can't do anything with your son, you have to stay at home, with your son, not doing anything, because if you put him in kindergarten, that will cost you a lot of money, a lot. And it's only for until 12 o'clock. What can I do with him the rest of the day, if I want

to work? So there, if you are a mother, you will not find many opportunities, working opportunities. Here ... I like it here. You can have your chance".

## 5. Discussion

Interestingly, neither gender nor interactions with gender predict the effect on the entrepreneurial intentions in the regression model. Indeed, the descriptive statistics confirm that highly-skilled women with refugee experience are equally likely (37.5 percent) as men (36.2 percent) to have entrepreneurial intentions, which is surprising when combined with two pieces of information. First, Syria is one of the countries with the lowest female self-employment rate in the world (Bastian and Zali 2016). Second, female respondents are characterised by a predominant lack of previous self-employment and their rather employment prone occupational choices. As expected, the share of women with previous self-employment experience (12.5 percent) is much smaller than the similar share of men (27.6 percent). Similarly, more women (15 percent) in comparison to men (10.3 percent) do not possess any previous professional experience. These two statistics, which shall have a disadvantageous effect on women's entrepreneurial intentions, may be compensated by several aspects.

First, similar to previous studies, entrepreneurial intentions are high for individuals with refugee experience, which usually does not translate into the actual rate of companies set up by refugees (Alexandre et al. 2019; Kachkar 2019; Mawson and Kasem 2019). The number of refugee-owned businesses is drastically lower than their declared intention, which shows that self-employment intention only is obviously not enough to start a business (Wauters and Lambrecht 2008). The high positive response rate of self-employment intention does not imply immediate action, but it may indicate some kind of readiness to act and adapt to a changing environment, which is an important feature in the case of newly arrived immigrants (Obschonka et al. 2018). Besides, research shows that high qualifications play a key role in declaring entrepreneurial intentions by individuals (Zybura et al. 2018).

Second, this readiness to act or adapt to a changing environment finds confirmation in the analysis of case studies. Both interviewed women said that they want to work in Sweden although they would not have worked back in Syria. An interesting aspect is that they explained such a significant shift in their refugee lives in two different ways. The first interviewee stressed the role of Swedish culture, according to which women are supposed to work. She explained that she would be bored and under risk of depression if she did not work in Sweden. In addition, the lack of sun and long periods of darkness would affect her mood negatively. The second interviewee highlighted the existence of women's rights in Sweden, which let her decide about her own life and pursue a professional career, which she had to stop back home the day she got married. Her case is an example of the emancipatory dimension of the refugee experience (Habib 2018). The flight from a country with a traditional division of gender roles to a democratic country with equality of women and men opens up opportunities for women with refugee experience (Habib 2018; Tahir 2020).

Third, both women acknowledge the role of childcare facilities. They benefit from access to public day care centres, which allows them to have time for their own self-development, for example, participation in Swedish classes. The theme of raising children emerged in both interviews. Although the interviewees have small children, both have displayed some entrepreneurial intention to start a company in Sweden. What is more, the first interviewee argued that self-employment helps to keep up work-life family balance. These findings confirm previous research emphasising the role of an encouraging environment, which to some extent diminishes the barriers women migrants may otherwise face in starting a business activity (Brieger and Gielnik 2020).

The regression analysis shows that previous self-employment predicts the effect on entrepreneurial intentions. However, the regression results examined together with descriptive statistics show that the previous self-employment variable works for highly-skilled men with refugee experience rather than women. The lack of previous self-employment

experience among highly-skilled women who want to open a company may be reconciled by the fact of having entrepreneurial models in their families (Wauters and Lambrecht 2006, 2008). Both interviewees had family members running their own businesses in Syria before the war broke out. As research points out, being surrounded by entrepreneurs increases the probability of having entrepreneurial intentions (Wauters and Lambrecht 2006). Second, as pointed out in both interviews, running one's own business is positively perceived in Syrian culture, which may be an additional stimulus for the non-experienced highly-skilled female respondent to declare entrepreneurial intentions (Vinogradov and Kolvereid 2007; Chand and Ghorbani 2011; Frederking 2004; Li 2001).

Also, the regression findings show that leadership aspiration predicts the effect on entrepreneurial intentions. Nonetheless, it is not clear to what extent it plays a role in the case of either highly-skilled men or women with refugee experience. Finally, our study shows that the occupational choices of 60 percent of female respondents, i.e., teachers, nurses and doctors, are the ones that should theoretically lead to dependent employment rather than self-employment (Zybura et al. 2018); however, the regression model does not confirm this trend. We only see from the descriptive statistics that the occupational choices that dominate among more than 50 percent of male respondents are in sectors that are much more prone to self-employment, i.e., sales, marketing, purchasing; administration, economics, law; building and construction; nature use and industry.

Last but not least, our empirical analysis helped us identify obstacles and enablers of entrepreneurial intentions, which often emerge in the context of individuals with refugee experience. In line with previous research, the obstacles to entrepreneurial intentions encompass a lack of previous self-employment experience and rather employment prone occupational choices (in the case of women), lack of start-up capital, lack of endorsement of business incubators and a necessity to know the host country language (Wauters and Lambrecht 2008). In terms of enablers of entrepreneurial intentions, we have identified the potential importance of entrepreneurial role models among family members, the positive perception of entrepreneurship in Syrian culture and the supportive role of the host country environment, i.e., gender-egalitarian host country and available childcare infrastructure. However, the latter findings definitely need to be analysed in a more consistent way while examining the macro institutional context of entrepreneurial intentions.

## 6. Conclusions

Our study contributes to the intersection of three strands of literature, i.e., entrepreneurial intentions, refugee migration experience and highly-skilled migrant women. The work analyses the individual level determinants of the entrepreneurial intentions of highly-skilled individuals with refugee experience with a particular focus on women. The empirical analysis embedded in the context of Sweden is two-fold. First, it is done in SPSS on the sample (*N* = 98) drawn from the 2017 Swedish Invandrarindex data set with the use of binary logistic regression. Second, the findings from the quantitative analysis are nuanced with the analysis of two case studies based on SSI with Syrian women with refugee experience.

In line with previous research, the findings confirm the significance of previous self-employment and leadership experience. The logistic regression model shows evidence of previous self-employment and leadership experience, but no gender and no employment prone occupation. The results show that highly-skilled women with refugee experience are equally likely to have entrepreneurial intentions as their male counterparts although, contrary to men, women in the study mostly did not possess previous self-employment experience. In addition, women's most common occupational choices, i.e., teacher or healthcare worker, are rather prone to dependent types of employment (Zybura et al. 2018). The entrepreneurial role models and the positive perception of entrepreneurship in Syrian culture observed in the case studies may also partially explain the high rate of entrepreneurial intentions among highly-skilled female respondents. Besides, the analysis of the case studies indicates the role of an encouraging environment for women to have entrepreneurial

intentions, which is the host country culture and children-dedicated institutions. One of the case studies points out the emancipatory dimension of refuge.

Finally, this study has some limitations. It focuses on one wave of refugees of a single nationality and only in the context of Sweden. The small sample size may suffer from low validity. We examine mostly individual level determinants of the entrepreneurial intentions of highly-skilled individuals with refugee experience. Thus, the meso and macro analysis encompassing the external context, institutional environment, policies, labour market structure, discrimination etc. is recommended across different host countries.

**Author Contributions:** Conceptualization, N.L.-B., B.G.; methodology, N.L.-B.; software, N.L.-B.; validation, N.L.-B.; formal analysis, N.L.-B.; investigation, N.L.-B.; resources, N.L.-B.; data curation, N.L.-B.; writing—original draft preparation, N.L.-B.; writing—review and editing, N.L.-B., B.G.; visualization, N.L.-B.; supervision, B.G.; project administration, N.L.-B.; funding acquisition, N.L.-B. Both authors have read and agreed to the published version of the manuscript.

**Funding:** This research was funded by NATIONAL SCIENCE CENTRE, grant number UMO-2018/31/N/HS4/02534.

**Institutional Review Board Statement:** Not applicable.

**Informed Consent Statement:** Not applicable.

**Data Availability Statement:** Restrictions apply to the availability of these data. Data was obtained from third party and are available at URL: http://www.invandrarindex.se/?lang=en with the permission of third party.

**Conflicts of Interest:** The authors declare no conflict of interest.

## Appendix A

(1) The data set Invandrarindex 2017 is not publicly available. It can be obtained upon request. The contact information is available at http://www.invandrarindex.se/?lang=en.

(2) Descriptive statistics. Educational background and work experience before coming to Sweden of highly-skilled respondents with refugee experience ($N = 98$).

**Table A1.** Educational background and work experience before coming to Sweden of highly-skilled respondents with refugee experience ($N = 98$).

| Variable Name | All | % | Women | % | Men | % |
|---|---|---|---|---|---|---|
| Educational background before coming to Sweden | | | | | | |
| Administration, economics, law | 21 | 21.4 | 6 | 15 | 15 | 25.9 |
| Building and construction | 4 | 4.1 | 1 | 2.5 | 3 | 5.2 |
| Data/IT | 10 | 10.2 | 2 | 5 | 8 | 13.8 |
| Industry | 4 | 4.1 | 1 | 2.5 | 3 | 5.2 |
| Transport | 1 | 1 | 0 | 0 | 1 | 1.7 |
| Nature use | 6 | 6.1 | 1 | 2.5 | 5 | 8.6 |
| Teaching | 26 | 26.5 | 15 | 37.5 | 11 | 19 |
| Health and hospital | 10 | 10.2 | 4 | 10 | 6 | 10.3 |
| Culture, media, design (e.g., journalists) | 3 | 3.1 | 2 | 5 | 1 | 1.7 |
| Natural science | 3 | 3.1 | 2 | 5 | 1 | 1.7 |
| Sales, purchasing, marketing | 1 | 1 | 0 | 0 | 1 | 1.7 |
| Hotel, restaurant, OOH | 1 | 1 | 0 | 0 | 1 | 1.7 |
| Other | 17 | 17.3 | 7 | 17.5 | 10 | 17.2 |

**Table A1.** *Cont.*

| Variable Name | All | % | Women | % | Men | % |
|---|---|---|---|---|---|---|
| Work experience before coming to Sweden | | | | | | |
| No experience | 12 | 12.2 | 6 | 15 | 6 | 10.3 |
| Administration, economics, law | 11 | 11.2 | 2 | 5 | 9 | 15.5 |
| Building and construction | 9 | 9.2 | 1 | 2.5 | 8 | 13.8 |
| Data/IT | 8 | 8.2 | 1 | 2.5 | 7 | 12.1 |
| Industry | 10 | 10.2 | 0 | 0 | 10 | 17.2 |
| Transport | 2 | 2 | 0 | 0 | 2 | 3.4 |
| Nature use | 10 | 10.2 | 1 | 2.5 | 9 | 15.5 |
| Teaching | 29 | 29.6 | 18 | 45 | 11 | 19 |
| Health and hospital | 8 | 8.2 | 4 | 10 | 4 | 6.9 |
| Culture, media, design (e.g., journalists) | 1 | 1 | 1 | 2.5 | 0 | 0 |
| Natural science work | 1 | 1 | 0 | 0 | 1 | 1.7 |
| Sales, purchasing, marketing | 12 | 12.2 | 0 | 0 | 12 | 20.7 |
| Hotel, restaurant, OOH | 3 | 3.1 | 0 | 0 | 3 | 5.2 |
| Other | 13 | 13.3 | 7 | 17.5 | 6 | 10.3 |

Note: The categories in both educational background and work experience are not exclusive.

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
