# Peer review of "What Determines the Entrepreneurial Intentions of Highly-Skilled Women with Refugee Experience? An Empirical Analysis in the Context of Sweden"

_admsci, doi:10.3390/admsci11010002_

Round 1

Reviewer 1 Report

This paper deals with a relevant issue and provides interesting evidence. My overall impression is that some points need to be additionally explained and some controls/extensions could help the evidence to become more solid:

1) I would suggest to extend theoretical analysis by focusing more on what determines the entrepreneurial intentions of highly-skilled women with refugee experience because now literature review section does not provide current understanding and background information about the topic explaining determinants of highly skilled women with refugee experience for the entrepreneurial intentions. and it is not clear how authors choose the variables which are included in the regression.

2) It is not clear the number of sample respondents because general sample is not described. it is only stated that N=98 and also qualitative analysis is based only on two case studies but authors do not give arguments for the validity of the sample.

3) How authors interpret the variable Gender x Age what is the meaning of that variable?

4) The interpretation of regression results is unclear what do you mean by self-employment unit?

5) the case studies is more like report without deeper interpretations.

Author Response

Thank you very much for your feedback. I have done my best to address all your comments in the improved version of the manuscript.

1) I have explained that there is no research yet on what determines the entrepreneurial intentions of highly-skilled women with refugee experience and this is the identfied research gap. I have also highlighted that I am using human capital theory and it determines which I have included in the regression.

96 – 109 in green

131 – 134 in green

142 - 146 in green

2) I have added the number of respondents in the data set (2526) and explained how I have selected the sample of interest (lines 238 and 265 - 273 in green). At the end I mention that "The small sample size may suffer from low validity."

596 in green

3) Hypothesis 5b (H5b). Younger highly-skilled women with refugee experience are less likely to have entrepreneurial intention than their male counterparts.

Hypothesis 5c (H5c). Younger highly-skilled women with refugee experience are less likely to have entrepreneurial intention than older highly-skilled men with refugee experience.

216 - 219 in green

4) I have corrected it.

354 and 361 in green

5) I have completely rewritten the analysis of case studies focusing on the theoretical concepts instead of narrating chronologically the stories of interviewees. I have divided this analysis into obstacles and enablers of entrepreneurial intentions.

367 - 497 in green

Reviewer 2 Report

The article is interesting and a good candidate for publication after some changes, some are quite challenging:

  1. The main issue with the article is the literature review. I strongly recommend an improvement. Methodologically it is difficult to understand the approach, but it looks like a hypothetical-deductive approach (predictor variables, quantitative findings, qualitative analysis...). However, in the LR there are no hypotheses or propositions to be presented. Thus, since the authors recognize that the study confirms some previous research (e.g. "The entrepreneurial role models and the positive perception of entrepreneurship in Syrian culture, observed in case studies, may also partially explain the high rate of entrepreneurial intentions among highly-skilled female respondents."), it would be better to articulate them in the LR around research hypothesis, which are confirmed to not in the studies.
  2. The authors present two studies, which is a very valuable strategy. However, the article could be more enriched if the two studies were better articulated in the discussion section.
  3. In the conclusions is missing the practical/policy-making/managerial implications.

Author Response

Thank you very much for your feedback. I have done my best to address all your comments in the improved version of the manuscript.

1) I have rewritten the literature review and highlighted the use of the human capital theory, which explains what kind of variables I use in the regression model. I have formulated in total hypotheses based on previous research.

2) I have rewritten the discussion and tried to discuss better the case studies together with regression results. I have changed the highlights and order of arguments in the discussion section.

503 - 505 in blue

511 - 520 in blue

541 - 544 in blue

553 - 555 in blue

562 - 572 in blue

3) I mention the need to conduct further research. In my opinion, making any policy recommnedations would be premature taking into consideration the early stage of research examining the intersection of refugee migration experience, gender and qualifications in the context of entrepreneurial intentions.

571 - 572 in blue

Reviewer 3 Report

The paper explores, analyses, and presents the individual-level determinants of entrepreneurial intentions of highly-skilled individuals with refugee experience, focusing on women. The idea underpinning the paper is refreshing and novel in my view and thus is highly commended. It also provides a good overview of entrepreneurship literature and makes a point that we need to facilitate the integration of refugees by encouraging them to set their own business. The choice of Syrian refugees in Sweden is also appropriate given that the country is well known for facilitating and helping those that seek asylum.

Nevertheless, the paper, in my view, could still be improved in a number of pertinent ways through well written.

First, The interviews showed that the refugee always has the right intention to start a business. However, they have several obstacles, like financial support and entrepreneurial education (coaching, consulting...). It would be better to add a short section about these obstacles and make specific recommendations in order to solve such an issue. You could also mention that in your conclusion.

Second, the element of entrepreneurship intention and the women entrepreneurship culture could be explained better. Attached to the manuscript, you will find some new references that could benefit the researcher. I would encourage the authors to review those papers and adopt useful ideas from there to support this articulation.

Finally, the paper does provide useful implications from its case study and analysis. Some small structural improvements would add better visibility to the manuscript (see attached comments).

I hope the authors will find these comments useful to improve the ideas herein this paper.

Good luck and all the best!

Author Response

Thank you very much for your feedback. I have done my best to address all your comments in the improved version of the manuscript.

1) First, according to your suggestion I have added a section about the obstacles and also enablers of entrepreneurial intentions.

402 - 457 in green Obstacles

458 - 497 in green Enablers

In my opinion it would be premature to make specific recommendations, so I prefer not to formulate them yet.

2) Thank you very much for the suggested references. They are valuable papers and I have cited them in my own paper.

58 in violet

61 in violet

Most of the references cited in the papers recommended by you are not available to me, so I had to resort to some other articles.

166 in violet

168 - 169 in violet

3) I have applied your comments to improve the structure of the manuscript.

Round 2

Reviewer 1 Report

All remarks were included in the revised manuscript

Reviewer 2 Report

I am satisfied with the improvements made. Congratulations to the authors.